# Platelet aggregates detected using quantitative phase imaging associate with COVID-19 severity

Christian Klenk [1,6], Johanna Erber[2,6], David Fresacher [1,3], Stefan Röhrl [3], Manuel Lengl[3], Dominik Heim [1], Hedwig Irl [4], Martin Schlegel [4], Bernhard Haller [5], Tobias Lahmer[2], Klaus Diepold[3], Sebastian Rasch [2,7] & Oliver Hayden [1,7 ✉]

## Abstract

**Background** The clinical spectrum of acute SARS-CoV-2 infection ranges from an asymptomatic to life-threatening disease. Considering the broad spectrum of severity, reliable biomarkers are required for early risk stratification and prediction of clinical outcomes. Despite numerous efforts, no COVID-19-specific biomarker has been established to guide further diagnostic or even therapeutic approaches, most likely due to insufficient validation, methodical complexity, or economic factors. COVID-19-associated coagulopathy is a hallmark of the disease and is mainly attributed to dysregulated immunothrombosis. This process describes an intricate interplay of platelets, innate immune cells, the coagulation cascade, and the vascular endothelium leading to both micro- and macrothrombotic complications. In this context, increased levels of immunothrombotic components, including platelet and platelet-leukocyte aggregates, have been described and linked to COVID-19 severity.

**Methods** Here, we describe a label-free quantitative phase imaging approach, allowing the identification of cell-aggregates and their components at single-cell resolution within 30 min, which prospectively qualifies the method as point-of-care (POC) testing.

**Results** We find a significant association between the severity of COVID-19 and the amount of platelet and platelet-leukocyte aggregates. Additionally, we observe a linkage between severity, aggregate composition, and size distribution of platelets in aggregates.

**Conclusions** This study presents a POC-compatible method for rapid quantitative analysis of blood cell aggregates in patients with COVID-19.

**Plain language summary**

The human body produces a series of immune responses when it gets infected with SARS-CoV-2, the virus that causes COVID-19. One of these responses involves platelets, the blood clotting factor sticking to immune cells to form cell aggregates in the bloodstream. We aimed to understand the significance of these cell aggregates in COVID-19 disease progression. A quantitative imaging approach was used to investigate the number and components of these cell aggregates in SARS-CoV-2 infected patient blood. We observed blood from severe COVID-19 patients was associated with higher numbers and specific composition of cell aggregates. Our method can potentially support the risk stratification of severe patients to prevent complications in COVID-19 and other medical disorders, where immune cells are shown to aggregate.

[1] Heinz-Nixdorf-Chair of Biomedical Electronics, School of Computation, Information and Technology, Technical University of Munich, TranslaTUM, 81675 Munich, Germany. [2] TUM School of Medicine and Health, Department of Clinical Medicine - Clinical Department for Internal Medicine II, University Medical Centre, Technical University of Munich, 81675 Munich, Germany. [3] Chair for Data Processing, School of Computation, Information and Technology, Technical University of Munich, 80333 Munich, Germany. [4] TUM School of Medicine and Health, Department of Clinical Medicine - Clinical Department of Anaesthesiology and Intensive Care Medicine, University Medical Centre, Technical University of Munich, 81675 Munich, Germany. [5] TUM School of Medicine and Health, Department of Clinical Medicine - Institute of AI and Informatics in Medicine, University Medical Centre, Technical University of Munich, 81675 Munich, Germany. [6] These authors contributed equally: Christian Klenk, Johanna Erber. [7] These authors jointly supervised this work: Sebastian Rasch, Oliver Hayden. ✉email: oliver.hayden@tum.de

The clinical manifestation of Coronavirus disease 2019 (COVID-19) caused by the severe acute respiratory syndrome coronavirus 2 (SARS-CoV-2) ranges from asymptomatic infection to a life-threatening disease, which is characterized by acute respiratory distress syndrome (ARDS) and multi-organ failure.

A hallmark of severe COVID-19 is COVID-19-associated coagulopathy (CAC) that can manifest in micro- and macro-thrombi leading to multi-organ injury and failure. While the underlying cellular and molecular mechanisms have not been completely understood, several lines of evidence suggest that CAC is driven by dysregulated immunothrombosis, a complex interplay of innate immune cells like neutrophils and platelets, the coagulation cascade, fibrinolytic pathways and the vascular endothelium[1–4]. Vascular endothelial cell dysfunction, a hyper-inflammatory immune response, and hypercoagulability result in a systemic prothrombotic phenotype in COVID-19 that has been linked to morbidity and mortality[4–7].

The clinical management of COVID-19 patients largely depends on disease severity and complications. Thus, early prediction of the course of the disease and markers for the occurrence of adverse events are highly relevant for patient management. Several attempts to establish predictive markers or prediction models have been made, and the investigated markers include numerous factors involved in CAC: Plasma levels of D-dimers are frequently elevated in COVID-19 patients and have been linked to thromboembolic complications, disease severity, and mortality[8]. Likewise, the C-reactive protein (CRP), another acute-phase protein and inflammatory marker, guides prognosis and disease severity and may also predict the risk of venous thromboembolism[9,10]. However, the relevance of both markers for clinical management, including anticoagulation, requires further investigation[9,11,12]. Numerous studies have evaluated platelet function in patients with COVID-19. There is growing evidence of platelet hyperactivation contributing to dysregulated immunothrombosis in COVID-19 via secretion of intracellular granula and surface expression of prothrombotic membrane proteins[4,7,13,14]. Several other markers of coagulation, fibrinolysis, endothelial dysfunction, extravesicular vesicles, and novel soluble biomarkers have been investigated[9]. While there is an evident potential for many of the markers to guide risk stratification, hardly any marker or algorithm on multiple markers has been translated into clinical practice. Reasons are diverse and include lack of biomarker accuracy, insufficient prospective validation, complex diagnostic workflow, and economics[15–17]. Specialized and not yet automated laboratory techniques are often required, rendering routine measurement in clinical practice unfeasible.

Further, increased levels of platelet aggregates (PP aggregates) as well as platelet-leukocyte aggregates (LP aggregates), including platelet-monocyte and platelet-neutrophil aggregates, have been described in COVID-19 patients, linked with disease severity, and could thus serve as biomarkers[7,18–23]. Those aggregates can be detected by several methods. One example is the usage of blood smear analysis, where PP aggregates and macroplatelets have been observed in COVID-19 patients[24]. However, the amount of blood cells that can be evaluated manually is limited and hampers statistical analysis. Fluorescence flow cytometry overcomes the problem of low statistical power but lacks spatial resolution and workflow automation. Forward and side scatter information combined with fluorescent antibodies staining enables cell detection with high specificity but lacks differentiation between single platelets and PP aggregates. Aggregates consisting of different cell types, such as LP aggregates, can be detected but not quantitatively evaluated. PP aggregates, however, cannot be analysed at all[23]. Also, the sample preparation steps for the fluorescent staining may interfere with cell biology and sample composition[25–29]. Image-based flow cytometer approaches combine the advantages of microscopy and flow cytometry and have shown promising results[23]. However, the potential negative effects from the time delay due to incubation periods and the possible interferences of other sample preparation steps remain. To our best knowledge, there is no label-free high-throughput methodology allowing the detection of aggregates and the analysis of their composition on a single platelet level suitable for point-of-care (POC) applications.

Here, we present a method that combines a quantitative phase imaging method, also referred to as digital holographic microscopy (DHM), with a microfluidic chip and a customized image analysis, resulting in a label-free high-throughput imaging flow cytometer (Fig. 1a). Contrary to most developments in the field of flow cytometry driven by throughput and multiplexing, we optimized our conditions to mimic the flow conditions in vessels at low shear stress of ~1.000 s$^{-1}$ (Supplementary Fig. 1a). The used imaging technique allows the acquisition of quantitative phase images with a suitable contrast. Since the light scattered by cells is significantly weaker than the incident light, phase information is more affected than the amplitude[30]. In this study, to the best of our knowledge, we provide the first evidence that quantitatively detecting blood cell aggregates in COVID-19 by label-free digital holographic microscopy can be a valid POC method to assess and predict the disease severity in critical care patients.

## Methods

**Study cohort**. 36 patients (age 32 to 83) with PCR-confirmed SARS-CoV-2 infection admitted to the Klinikum München rechts der Isar between November 2020 and 2021 were prospectively enrolled. Blood samples (arterial or venous) were collected using 2 ml Blood Gas-Monovettes® from Sarstedt and subjected to DHM analysis within 30 min after the blood draw. To minimize mechanical stimuli, samples were carefully transported to the nearby prototype. Demographic, clinical, and laboratory data were retrospectively compiled by chart review. Based on the *WHO ordinal scale for clinical improvement for hospitalised patients with COVID-19*[31], patients were classified into moderate COVID-19 (score 4 or 5) and severe COVID-19 (score 6 to 10). Female and male volunteers (age 29 to 67) served as blood donors for reference measurements. The study was conducted in accordance with the declaration of Helsinki and approved by the ethics committee of the Technical University Hospital of Munich (221/20 S-SR and 620/21 S-KK). Informed consent was obtained from all subjects or their legal representatives.

**Sample preparation**. Blood samples were diluted in a measurement buffer with a factor of 1:100 (10 μl blood, 990 μl buffer) and immediately subjected to analysis. The measurement buffer contains 99.95% phosphate-buffered saline (PBS) and 0.05% polyethylene oxide (PEO, molecular weight $M_W = 4 \times 10^6$ Da, Sigma Aldrich).

**Microfluidic cell alignment**. Microfluidic PMMA chips with a channel diameter of 500 μm, a height of 50 μm, and a length of 5000 μm were used for high throughput measurements with precise blood cell alignment in a sub-monolayer. To achieve this, viscoelastic and hydrodynamic focusing methods were combined to ensure >90% of blood cells were in focus (Supplementary Fig. 1b). For hydrodynamic focusing, a channel containing 5 inlets and one outlet was used (Fraunhofer IMM). Only one inlet serves as a sample inlet while the other four serve as top, bottom, and side sheaths surrounding the sample. This enables the alignment of the blood cells in horizontal and vertical directions. To support the focusing of platelets in the z-direction, 0.05% 2 MDa PEO is added

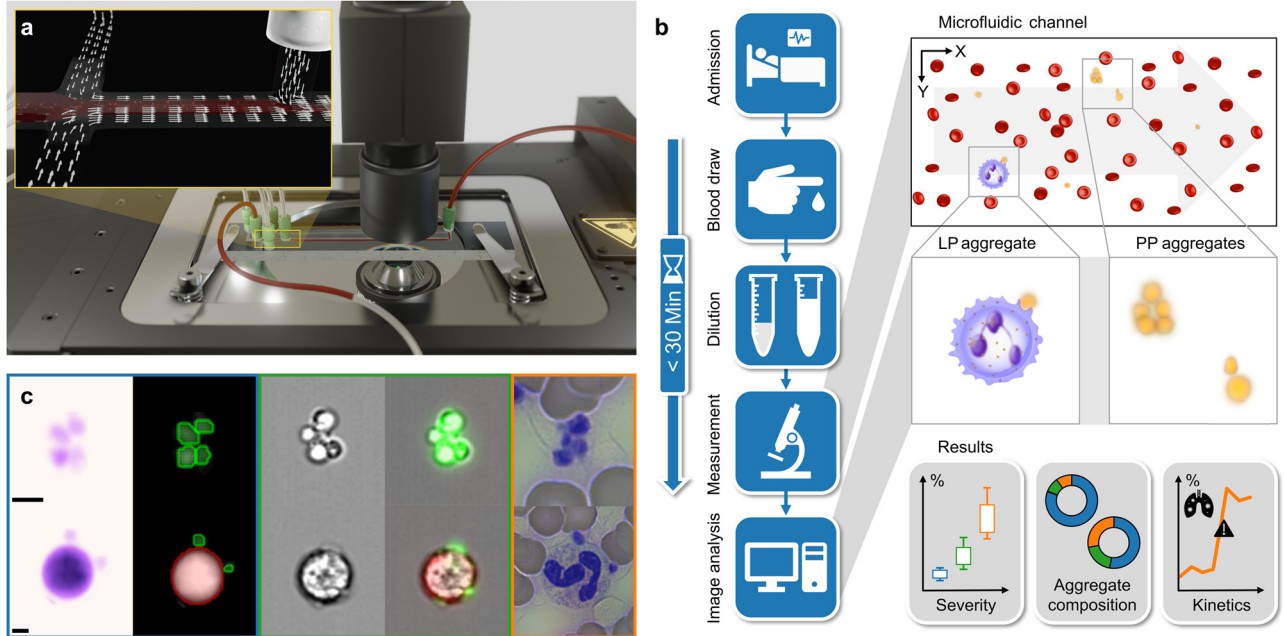

**Fig. 1 Novel image-based flow cytometry enables the identification of cell aggregates and their components. a** Scheme of a flow cytometry setup including a DHM and a microfluidic chip. The insert illustrates the applied principle of hydrodynamic focusing in two dimensions. **b** Study workflow: consisting of admission of patients to the hospital, blood draw, sample dilution, measurement in the microfluidic channel (within 30 min from collection), and image analysis. The measurement time is 2 min. The number and composition of the aggregates were correlated with the severity and courses of COVID-19 patients. **c** Platelet aggregates (top row, PP) and leukocyte-platelet aggregates (bottom row, LP) were captured with different imaging techniques as reference. From left to right, exemplary images detected by DHM in false colour, DHM images with single cells detected by the algorithm (blue box), brightfield images with an AMNIS imaging flow cytometer, AMNIS images of brightfield merged with the fluorescence images (green box), and images of a blood smear (orange box) are depicted. The green circles (DHM images) indicate that the algorithm detects a platelet, whereas green-fluorescent cells (AMNIS images) represent CD61-labeled cells. Likewise, red circles represent leukocytes detected by DHM, while CD45-labeled cells are red fluorescent. Both scale bars correspond to 5 μm.

to the solution to enable viscoelastic focusing. Preliminary tests were carried out to ensure that the addition of the polymer did not activate the platelets (Supplementary Fig. 2c, d). Furthermore, the flow rate was adjusted to minimize the shear rate of $\sim 1.000\ \text{s}^{-1}$ acting on the aggregates (Supplementary Fig. 1a). With a total flow rate in the channel of 1.6 μl/s, image acquisition was performed with a flow velocity of 6.4 cm/s. The bottom and top sheaths are both set to 0.2 μl/s, whereas the side sheaths are maintained at 0.5 μl/s. Laminar flow with maximum Reynolds numbers in the single-digit range (Re ≈ 6) are achieved in this condition.

**Digital holographic microscopy.** Measurements were performed with a transmission digital holographic microscope (Ovizio Imaging Systems). Interferograms are generated by off-axis holography combined with a double-shearing interferometric approach. Partially coherent light is emitted by a 528 nm SLED (Osram) in combination with a Koehler illumination unit and passes through the sample toward the objective. The objective (CFI LWD, Nikon) has a 40× magnification and a numeric aperture of 0.55 for a compromise between lateral and axial resolution to ensure a high yield of platelets in focus. A grating is used to generate the different diffraction maxima, which, together with a spatial filter, lay the foundation for the double-shearing approach. Interference of the single beams is recorded with the camera at an off-axis angle between the beams, resulting in the interferogram. The GS3-U3-32S4M camera (Teledyne FLIR LLC) generates 105 images per second at an exposure time of 5 μs. Reconstruction of the phase and amplitude images out of the holograms was performed using the OsOne Software Version 5.12.12 (Ovizio Imaging Systems). Parallel measurement of up to

20 cells per image allows high-throughput of 500−2000 cells/s. For each test, ~50,000 blood cells per captured for analysis.

**Image analysis.** The Image Analysis Pipeline is composed of three distinct steps: preprocessing of the phase images, Mask R-CNN based segmentation, and analysis of detected cells and aggregates.

Preprocessing is the first step of the image analysis pipeline, which includes background subtraction, cell detection, masking, and normalization. Background Subtraction is needed to remove background noise and artifacts of the fluid channel. Cell Detection is enabled by thresholding and contour finding in the acquired images. In the first step, a binary threshold was applied to the phase images. Contours were extracted from the resulting binary images based on the algorithm of Suzuki et al. [32]. Subsequently, these contours were filtered based on size. Each cell or cell aggregate (represented by a contour) was then saved as a smaller image section for further processing. Masking of individual cells was used to remove unwanted noise from fluid, particles, and other cells. This was done based on the earlier calculated threshold mask. Normalization is the last essential preprocessing step. The images were first clipped to limit the value range. Afterward, min–max normalization was applied to transform the image values into the range 0–1 suitable for neural networks.

Segmentation is the second step of the image analysis, which allows identification of single blood cells, aggregates, and their components. In this work, we employ a combination of classical thresholding for a rough evaluation of the cell locations for preprocessing and, as a second stage Mask R-CNN[33] for refining the individual aggregate components and identifying their cell

types. Mask R-CNN allows both object detection and object mask computation simultaneously by performing four steps. In the first stage, a convolutional network (backbone) receives the input images and provides a convolutional feature map. Secondly, a region proposal network (RPN) provides candidate regions called regions of interest (RoI). This is done by sliding a small network over the convolutional feature map. In the third stage, a RoI align layer utilizes bilinear interpolation to provide feature maps of the same size as the RoI. These maps are then used in the fourth stage for classification and bounding box regression. A small FCN is applied to each RoI to predict the individual object masks on a pixel resolution. In this work, a ResNet50[34] is used as a backbone.

The performance and reliability of a neural network depend highly on the quality, amount, and selection of training data. For this purpose, a great amount of ground-truth annotated data is needed. However, manual labeling of cell images is, in this case, not feasible due to the quality and quantity of required ground-truth data. Therefore, we used a method of creating a synthetic dataset of aggregates, made by composing multiple classified single-cell images together to form cell aggregates. This is based on pure blood cell populations extracted from whole blood.

Using this synthetic dataset of aggregates, the neural network is trained on 200,000 cell images in total. The performance of the trained network is evaluated on both computer-generated aggregates as well as manually labeled images.

Analysis of the detected cells and aggregates is the last step of the processing pipeline. Hereby, based on the segmentation and classification results, image patches are categorized as single cells or cell aggregates. The number of cells in an aggregate as well as their cell classes and their morphological parameters are saved for each patch. This allows an in-depth analysis of both single components and whole aggregates at the same time.

All image analysis tools were written in Python 3.8.8 (Python Software Foundation). Computer vision algorithms were mainly adapted from OpenCV 4.5.5[35], and the neural network components were built in Tensorflow 2.8.0[36].

**Statistics and reproducibility**. Statistical analyses were performed using IBM SPSS Statistics 25 (SPSS Inc), the statistical software R version 4.0.2 (R Foundation for Statistical Computing), Microsoft Excel (V16.51), and OriginPro (2021b). The Shapiro-Wilk test was applied to test for normal distribution. The significance of non-normal distributed data was analysed using the non-parametric Kruskal-Wallis-ANOVA with a subsequent Dunn's test where significance levels are indicated by asterisks (*$p < 0.05$; **$p < 0.01$; ***$p < 0.001$ and ****$p < 0.0001$). Correlation values between aggregates and various clinical parameters were obtained by Spearman's rank correlation. To consider multiple observations per patient, within- and between-subject correlations between platelet aggregates and relevant biochemical parameters were estimated. Linear and exponential growth function fits were executed using OriginPro (2021b). Missing data were not imputed. Graphs were plotted using OriginPro (2021b) and R studio (1.4.1106) and assembled using Inkscape (1.1.2).

Patients ($n = 36$) and healthy references ($n = 15$) were measured once at each sampling. Longitudinal samples (maximum 5, interval 2 to 5 days) were intended to be collected, starting as early as possible from admission to the ICU. Each measurement contains approximately 50,000 blood cells, which were recorded in less than two minutes.

**Reporting summary**. Further information on research design is available in the Nature Portfolio Reporting Summary linked to this article.

## Results

**Identification of blood cell aggregates**. We sought to identify and characterize blood cell aggregates of low-contrast platelets using a high-throughput imaging flow cytometer with gentle microfluidic guiding[37,38]. The technical details are described in the method section, and a depiction of the cytometer setup is presented in Fig. 1a. In brief, we keep the time from sample collection to analysis to less than 30 min (Fig. 1b) and confine the preanalytical sample preparation to a single dilution step, as platelets can be artificially activated by external influences such as storage times and mechanical stimuli[25–29]. The method was optimized to minimize both shear rates and spatial coincidences. Sub-monolayers of blood cells are measured in a microfluidic channel (Supplementary Fig. 3) using a transmission DHM. Image analysis was enhanced to determine the number of single cells (erythrocytes, platelets, and subtypes of leukocytes) and the composition of cell aggregates (Fig. 1c). This label-free, straight-forward workflow predestines our method to be established as a POC diagnostic tool.

We then employed our prototype to study cell aggregate formation in COVID-19 using peripheral blood from patients with a positive SARS-CoV-2 polymerase chain reaction test (PCR) ($n = 36$) and blood from healthy donors as control ($n = 15$). Based on the WHO ordinal scale for clinical improvement for hospitalized patients with COVID-19, the patient cohort was divided into a moderate (score 4 or 5) and severe group (score 6 to 10)[31]. The patient characteristics are described in Supplementary Table 1, Supplementary Fig. 4 and Supplementary Data 1. To get a deeper understanding of aggregate kinetics and its correlation to disease severity, samples from COVID-19 patients were longitudinally collected, resulting in a total of 118 independent measurements.

**PP aggregates and COVID-19 severity**. First, we compared the distribution of maximum blood cell aggregate levels of each patient between groups of disease severity. We found that platelet (PP) aggregate formation was significantly higher in both moderate and severe COVID-19 patients compared to healthy donors (Fig. 2a). Of note, the amount of PP aggregates was higher in severe COVID-19 compared with moderate disease, indicating that PP aggregate formation correlates with COVID-19 severity. The mean value of PP aggregates ranged from 1.52% for healthy, 4.00% for moderate, to 22.71% for severe. In line with this, we found significantly more PP aggregates in fatal COVID-19 cases (mean = 25.60%) compared to survivors (mean = 13.76%, Fig. 2b). Second, PP aggregates were more frequent for both survivors and non-survivors compared to references (mean = 1.52%). Third, leukocyte-platelet (LP) aggregates were significantly more frequent in severe COVID-19 compared to both healthy donors and moderate COVID-19 (Supplementary Fig. 5a). Last, maximum LP aggregate levels did not significantly differ between healthy donors and moderate disease.

To test whether cell aggregates qualify to predict severity in COVID-19 patients, we then analysed the cell aggregates detected at the initial measurement (day 0). PP aggregates were significantly higher in severe (9.06%) and moderate COVID-19 (3.29%) patients compared to healthy donors (1.52%). In line with the maximum level, PP aggregates were higher in severe COVID-19 compared to moderate cases at day 0 (Fig. 2c). Likewise, LP aggregates were significantly increased in severe COVID-19 compared to healthy individuals. LP aggregates were also enriched in severe COVID-19 compared to patients with moderate disease. However, the difference was not significant. Regarding LP aggregates, we did not find a difference between

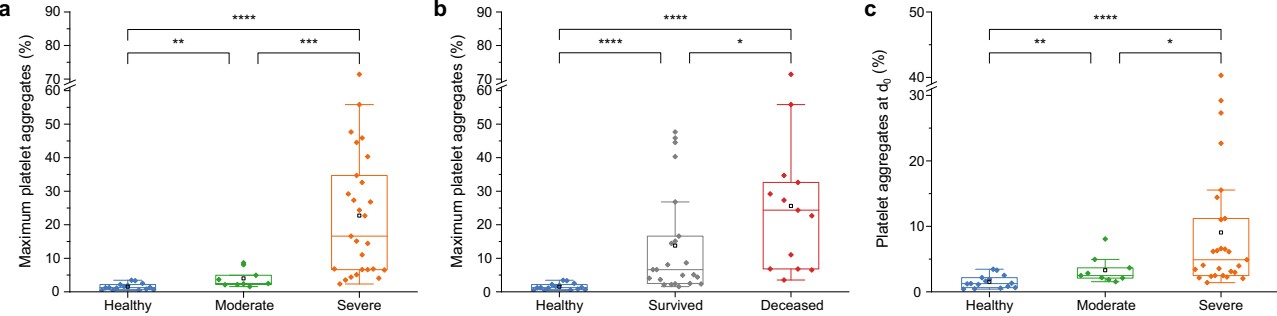

**Fig. 2 Influence of COVID-19 severity on platelet aggregate formation.** The distribution of PP aggregates is visualised by boxplots for healthy references (HR, $n = 15$), moderate ($n = 9$), and severe ($n = 27$) COVID-19 patients (**a**, **c**) and for HR ($n = 15$) compared to COVID-19 survivors ($n = 23$) and fatal cases ($n = 13$) (**b**). Each data point represents one patient. The centreline of the boxplots represents the median, and the framed black square is the mean of each distribution. The bottom and top borders of the box correspond to the first and third quartiles. Significance was tested by a two-sided Kruskal-Wallis ANOVA with Dunn's test and is indicated by asterisks. **a** Comparing the maximum PP aggregates measured during the study period, a significant difference can be observed between healthy vs. moderate ($p = 0.0046$), moderate vs. severe ($p = 0.0006$), and healthy vs. severe ($p < 0.0001$). **b** The percentage of PP aggregates on the day with the highest amount of PP aggregates is significantly higher in fatal COVID-19 cases compared to survivors ($p = 0.0285$) and both survivors or non-survivors compared to healthy references ($p < 0.0001$). **c** At day 0, PP aggregates are significantly higher in COVID-19 patients compared to healthy references (severe vs. healthy, $p < 0.0001$, moderate vs. healthy, $p = 0.0095$) and in severe COVID-19 patients compared to moderate cases ($p = 0.0326$).

moderate disease and healthy donors at the initial measurement (Supplementary Fig. 5b).

**PP aggregates and SARS-CoV-2 variants**. We then compared PP aggregates in patients infected with wild-type SARS-CoV-2 ($n = 10$) to patients infected with variant SARS-CoV-2 ($n = 26$), which included the B.1.1.7 (alpha, $n = 4$), B.1.351 (beta, $n = 1$) and B.1.617.2 (delta, $n = 21$) variants (Fig. 3a). PP aggregates at day 0 were significantly higher in patients infected with a SARS-CoV-2 variant (mean 11.80%) compared to wild-type SARS-CoV-2 (mean 3.57%). Again, PP aggregate levels on day 0 were significantly lower in healthy donors (mean 1.52%) compared to both the wild-type and variant cohorts.

**Correlation with clinical biomarkers**. We next aimed to correlate our findings with established biochemical parameters. Elevated D-dimer levels have been associated with coagulopathy and suggested as a biomarker to predict mortality in COVID-19[39]. Plotting longitudinally assessed PP aggregates with corresponding D-dimers, we found a moderate but significant between-subject correlation between PP aggregates and D-Dimer levels, indicating that D-dimer levels tend to be high in patients with high levels of PP aggregates (Fig. 3b). The within-subject correlation was not significant ($r = 0.24$; $p = 0.0920$). Further, we found a significant within-subject correlation between PP aggregates and procalcitonin (PCT), which is an established marker for hyperinflammation and used to monitor bacterial (super)infections in clinical practice (Fig. 3c). This suggests that dynamic changes may be monitored using serial measurements of PP aggregates. Our data did not indicate a positive between-subject correlation between PP aggregates and PCT levels ($r = 0.30$; $p = 0.1234$). While there is an association of CRP with severity in our cohort (severe vs. moderate COVID-19: 14.83 mg/dl vs. 7.53 mg/dl, $p = 0.05$), we did not observe a correlation between PP aggregates and the inflammatory markers CRP and Interleukin-6.

**Confounding factors**. A delay in measuring unfixed blood samples resulting from pre-analytical steps, incubation, or other storage times is known to activate blood platelets[25], which can lead to artificial platelet aggregation. To investigate these alterations, we obtained blood samples from nine severe COVID-19 patients and three healthy references and assessed PP aggregate percentages in each sample after 0, 60, and 120 min. We found a significant difference between both groups right after sample collection with mean values of 1.69% and 8.91% for healthy and severe, respectively. With increasing storage time, the number of aggregates and standard deviation increases in both groups impeding the differentiation of the two clusters at 120 min (Fig. 4a (16.01% vs. 22.81%, respectively). Most hospitalized COVID-19 patients receive antithrombotic therapy. The dosages may vary depending on prevalent thromboembolism, individual risk factors, underlying conditions, and local recommendations, aiming for thromboprophylaxis to therapeutic anticoagulation. In our cohort, 32 of 36 COVID-19 patients received unfractionated or low molecular weight heparin at least once during admission to the ICU. To exclude the effect of intravenous heparin on aggregate stability, we assessed the anticoagulation administered at the time of the blood sample collection. As depicted in Fig. 4b, we could not find a significant correlation between the dose of unfractionated heparin and the percentage of PP aggregates.

**Insights into the aggregate composition**. In addition to determining aggregate concentration, the described phase imaging method enables resolving the low-contrast platelets and therefore an evaluation of the aggregate composition. We found that the number of platelets forming an aggregate differs in COVID-19 patients, depending on disease severity (Fig. 4c, d and Supplementary Fig. 6). In healthy references, on average, 81.3% of the detected PP aggregates consisted of two platelets, compared to 73.2% and 59.8% in moderate and severe COVID-19, respectively. Consequently, the proportion of aggregates consisting of triplets or more platelets was higher. The mean proportion of aggregates composed of three platelets was 14.6% in healthy donors, 15.9% in moderate, and 26.6% in severe COVID-19 patients. In line with this, circulating aggregates with four platelets were found in 2.1%, 4.3%, and 8.7% of healthy donors, moderate and severe COVID-19, respectively. In severe disease, rare aggregates of up to 10 platelets were detected (Fig. 4c, Supplementary Fig. 6). We further aimed to investigate the size distribution variations of platelets within the micro-aggregates. We compared the size distributions of singular and aggregated platelets in our cohorts. We revealed that the proportion of larger platelets is higher in PP aggregates and increases with COVID-19 severity (Supplementary Fig. 7).

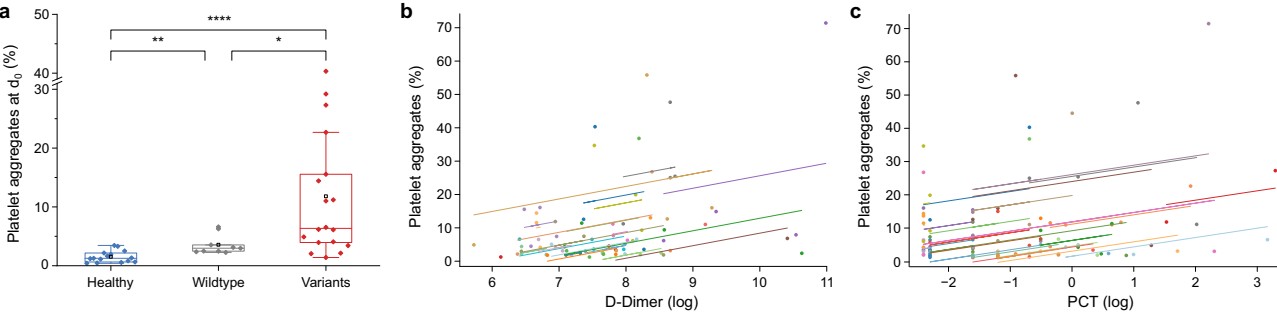

**Fig. 3 Correlation of PP aggregates with SARS-CoV-2 variants, D-Dimer and PCT. a** Comparison of the PP aggregates (in percentage at day 0 on the ICU) between healthy references and COVID-19 patients infected with wild-type SARS-CoV-2 or variants (B.1.1.7, B.1.617.2, with AY lineages and B.1.351). A significant difference was observed between wild-type vs. variants ($p = 0.0397$), wild-type vs. healthy ($p = 0.0021$), and variants vs. healthy ($p < 0.0001$). **b** Correlation of PP aggregate percentages with the D-Dimer (in µg/l, logarithmical, reference <500 µg/l) for all available measurements (maximum 4 per patient). Each colour represents one patient. Patients with high aggregate levels tended to have high D-dimer levels (between-subject correlation: $r = 0.46$; $p = 0.0131$). **c** Correlation of PP aggregate percentage with procalcitonin (PCT, in ng/ml, logarithmical, reference <0.1). Each colour represents one patient. PP aggregates and PCT levels correlate over the course of the disease (within-subject correlation: $r = 0.26$; $p = 0.0279$, measurements 1–4).

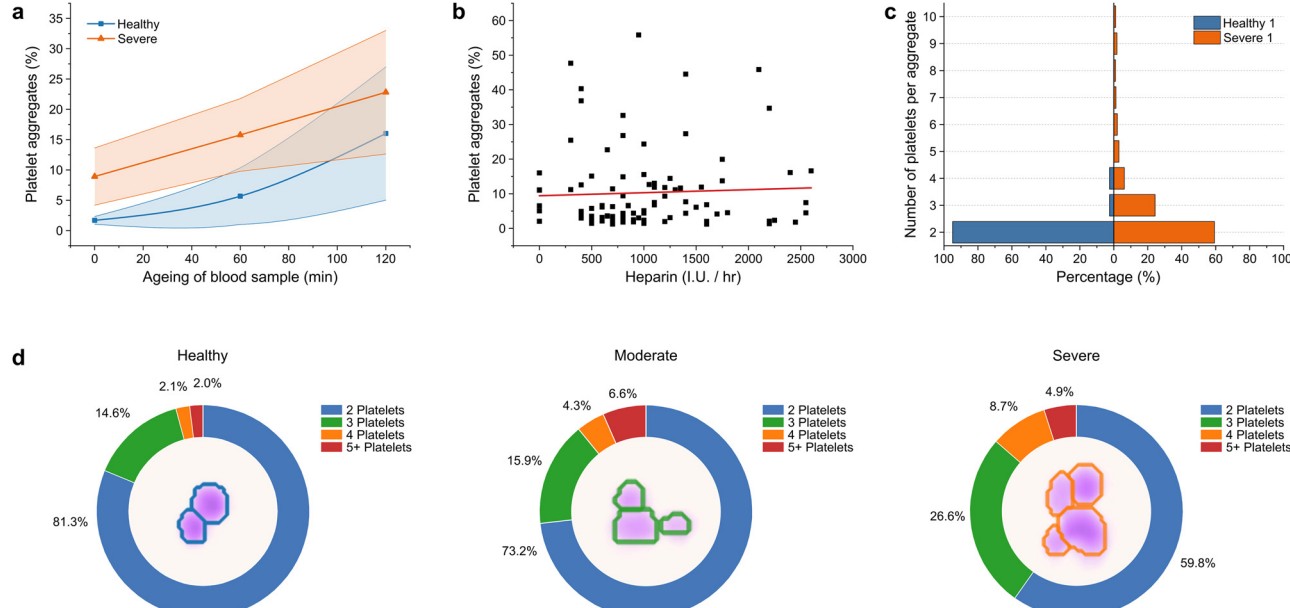

**Fig. 4 Confounders and composition of PP aggregates. a** Blood samples were repetitively measured to study ageing effects in heparin blood tubes. PP aggregate shares at minutes 0, 60, and 120 are shown for healthy donors (blue) and severe COVID-19 patients (orange). Squares and triangles show the mean values and the coloured area indicates the standard deviation. For 0 and 60 min, a significant difference between healthy donors and severe COVID-19 patients was observed ($p = 0.0074$ and $p = 0.0185$, respectively). In contrast, there is no longer a significant difference at 120 min between both groups ($p = 0.2911$). **b** No significant correlation between PP aggregate percentages and the intravenous unfractionated heparin dosage administered at the time of the blood sample collection for all measurements ($n = 98$) performed in patients ($n = 27$) receiving unfractionated heparin ($r = 0.05$; $p = 0.6268$). **c** Exemplary size distribution of platelet aggregates for a healthy donor (blue) and a severe COVID-19 patient (orange). **d** Comparison of the size distribution of PP aggregates in healthy donors, moderate and severe COVID-19 patients (maximum level). The shares represent the mean value of all patients from the respective collective. Images in the centres show aggregates containing two, three, or four platelets and their segmentation (coloured lines).

**Longitudinal measurements.** In hospitalized COVID-19 patients, no reliable biomarker for clinical deterioration has been established to date. We therefore aimed to longitudinally measure PP aggregates by label-free high-throughput phase imaging to investigate whether aggregate formation changes during the acute phase of the COVID-19 disease. We exemplarily show PP aggregates, PP aggregate composition, and the total platelet count as measured with a commercial haematology analyser (Fig. 5a) and representative computed tomography (CT) scans (Fig. 5b) for two severe COVID-19 patients over the course

of 11 and 20 days, respectively. Patient 1 was admitted to the ICU on day 0 due to an increasing need for oxygen. The patient stabilized and could be transferred to the normal ward on day 5, invasive or non-invasive ventilation was not required. The total platelet count of this patient slightly increased during the first week of ICU admission. Still, PP aggregate formation was almost stable below 5%, although aggregate size marginally increased on day 11[18]. The CT scan showed typical, moderate ground glass opacities. In contrast, patient 2 required intubation and mechanical ventilation nine days after ICU admission and

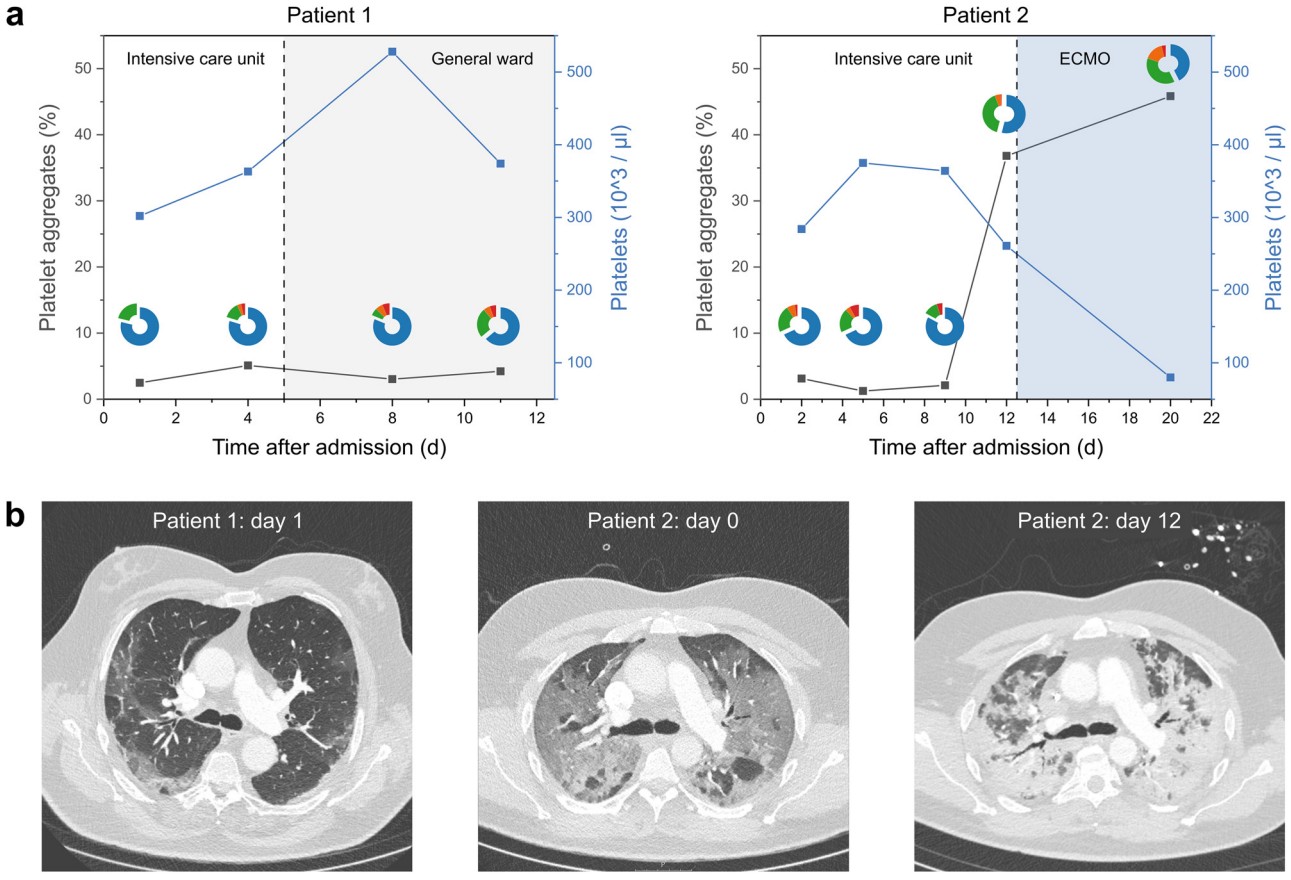

**Fig. 5 Longitudinal assessment of PP aggregates in two exemplary patients during hospitalization. a** The total platelet count and PP aggregates share for longitudinal measurements are shown as blue and black squares. Pie charts indicate the fractions of aggregates consisting of two (blue), three (green), four (orange), or more than four (red) platelets. Clinical events are highlighted by coloured areas. **b** Representative images of computed tomography (CT) scans from patient 1 the day after admission to the ICU (day 1, left) and from patient 2 at the admission to the ICU (day 0, middle) and the day before initiation of extracorporeal membrane oxygenation (ECMO) therapy (day 12, right).

venovenous extracorporeal membrane oxygenation (ECMO) 12 days after ICU admission due to a rapid deterioration of the respiratory function. While the total platelet count was within the reference range and no increase of PP aggregates was observed during the first 9 days, there was a significant increase in PP aggregates with a large fraction of aggregates containing three or more platelets on day 12, right before initiation of ECMO therapy. Due to the clinical deterioration on that day, a second CT scan showed dramatic progression compared to the initial scan (Fig. 5b). Following the initiation of ECMO therapy, the total platelet count markedly decreased. Intriguingly, and in line with the further clinical deterioration of the patient, the number of platelet aggregates and the size of the aggregates markedly increased on day 20. Unfortunately, we could not analyse blood samples hereafter due to protocol limitations. However, the patient gradually recovered, was weaned and subsequently removed from ECMO and ventilation, and could finally be discharged.

## Discussion

An unprecedented research effort has been initiated to understand CAC that is seen as a key pathogenetic factor and life-threatening complication in severe COVID-19[4]. Still, many pathophysiological mechanisms as well as anti-immunothrombotic and therapeutic approaches remain elusive. Biomarkers are important in the guidance of risk stratification, prognosis, and therapy. However, only three biomarkers that had been routinely measured before COVID-19 have recently been recommended in a consensus statement for routine use. The International COVID-19 Thrombosis Biomarkers Colloquium pronounced a good level of evidence for the platelet count, D-dimer levels, and CRP[9]. Other investigated biomarkers originating from CAC and linked to disease severity have not been recommended for clinical practice yet. Most markers emerged from observational, retrospective studies and lack prospective validation. Further, many biomarkers were only assessed once during the disease, limiting the evaluation of dynamic changes. Moreover, the optimal cut-off level often remains unclear. An underlying problem is the laboratory expertise and complexity of the assays required to assess the novel biomarkers, which hampers serial measurements, large-scale validation studies, and the use in clinical practice[9]. Readily accessible, quick, and inexpensive testing strategies, preferably POC diagnostic tools, are thus required to overcome this problem for acute care.

We here aimed to establish a method that enables high-through measurement of cell aggregate concentration and their composition by phase imaging flow cytometry. Using DHM allows for the label-free differentiation of blood cells due to the sufficient contrast of the corresponding phase images. This benefit enables measurement without time-consuming sample preparation but comes at the expense of losing specificity. Like other microscopical techniques, this method is limited by the numerical aperture of its objective lens[30]. In previous studies, DHM was used to analyse changes in T-cell morphologies in response to bacterial challenges[40], investigate different cellular patterns in

leukaemia patients[37] and detect viral SARS-CoV-2 particles[41]. Contrary to high-throughput flow cytometry, we perform micro-aggregate measurements at low shear rates mimicking blood vessel flow conditions[42] and still achieve high throughput of 500 - 2000 cells/s. Limiting the time from blood draw to measurement to less than 30 min, we secure the quality of the blood samples and avoid sample ageing effects leading to aggregate artifacts. This allows us to resolve not only the concentration of PP and LP aggregates but also the composition of PP aggregates and the size distribution of the contributing platelets.

To our best knowledge, this is the first time that this technique was applied to study blood cell aggregates in COVID-19. We show that PP aggregates are significantly associated with COVID-19 severity and mortality and thereby establish a potential POC tool to predict the severity of COVID-19. Our results are in line with recently published observations[4,13,21–23]. For example, Nishikawa et al. described an association between PP aggregate levels and the mortality[23]. While only the highest concentration of platelets was associated with clinical outcomes in this study[7,23], our data robustly showed an association of platelet aggregates and disease severity at the initial measurement and for the maximum level. Further, blood samples were undertaken with extensive preparation prior to analysis, which was performed within 4 h from sample collection[23]. In contrast, our findings suggest that a prompt measurement of the blood sample is required to prevent sample ageing effects (Fig. 3a), including increased variation and loss of micro-aggregate resolution. However, it should be noted that ageing effects may vary depending on the blood collection tube and containing anticoagulant. In our study, commercial blood gas monovettes pre-filled with calcium balanced heparin were used. While the effect resulting from the pre-filled heparin should be equal to all samples and outweigh the concentrations of systemically applied heparin dosages, we still aimed to investigate whether systemic heparin treatment of the patients affected aggregate formation. We could not find a correlation between the dose of unfractionated heparin and PP aggregates in our cohort. We note that nine patients were on regular treatment with acetylsalicylic acid (100 mg once per day), which irreversibly inhibits the cyclooxygenase-1 enzyme and reduces both arterial and venous thromboembolic events. In our study, we could not observe a correlation between the acetylsalicylic acid treatment and the amount of PP aggregates in the patient's blood (Supplementary Fig. 8c). Comparable to our findings, acetylsalicylic acid treatment did not reduce the mortality or risk of progression to invasive mechanical ventilation in a large, randomized controlled trial. However, thrombotic events were slightly decreased in patients treated with acetylsalicylic acid (150 mg once per day)[43].

In addition to an increased fraction of PP aggregates, we show that the aggregate composition varies with severity. Circulating PP aggregates of healthy donors mainly comprise two platelets (>80%). It is unlikely that an overlap of the cells solely explains the occurrence, as the percentage of platelets in whole blood compared to erythrocytes is low, and the dilution minimizes spatial coincidences. PP doublets may thus appear physiologically or as a side product of the blood draw (Supplementary Figs. 2a, 3). Patients suffering from moderate and severe COVID-19 tended to have larger PP aggregates consisting of up to ten platelets, including large platelets (Fig. 4, Supplementary Figs. 6, 7). This is in line with an early report of Rampotas et al., who observed large platelet aggregates of up to 30 platelets in blood films from COVID-19 patients treated on the ICU[24]. However, the role of the composition of PP aggregates and large platelets and the prognostic value for disease severity remains to be determined.

Our data indicate that PP aggregates are significantly higher in patients infected with SARS-CoV-2 variants than wild-type

SARS-CoV-2. In our cohort, molecular analysis revealed the B.1.617.2 (Delta) variant in 21 and the B.1.1.7 (alpha) variant in 4 of 26 patients infected with SARS-CoV-2-variants. Both variants are known to be more virulent compared to wild-type SARS-CoV-2[44,45]. We assume that the observed increase of PP aggregates mirrors disease severity and cannot be attributed to specific variants.

Our cohort size limited the correlation of PP aggregates with specific clinical outcomes, including thromboembolic events. For the same reason, the observed associations of CRP serum concentration and PP aggregates with COVID-19 severity did not translate into a correlation between the two parameters. However, we correlated two routinely assessed laboratory parameters with serial measurements of PP aggregates. D-Dimers result from degradation of fibrin in thrombi, are used as markers for coagulation and fibrinolysis, and have strongly been linked with the severity of COVID-19[8]. However, optimal cut-off levels for diagnostic or therapeutic decisions and the value of D-Dimer in disease monitoring remain a matter of debate[46]. In line with a prognostic value for disease severity, we observed a positive between-subject correlation for D-Dimers and PP aggregates. In contrast and consistent with previous reports, longitudinal measurements of D-Dimer and PP aggregate levels did not correlate[23]. The prognostic value of PCT in COVID-19 is equally conflicting[47,48]. Currently, serial assessment of PCT levels is recommended to identify secondary bacterial infections that often require antibiotic treatment[49]. Intriguingly, we found a significant within-subject correlation between PP aggregates and corresponding PCT levels ($r = 0.26$; $p = 0.028$), hinting at a possible link of PP aggregates with superinfections that might be interesting to be addressed in future projects.

We finally present two exemplary cases showing a correlation between the serial assessment of PP aggregate levels and the clinical course. While PP aggregates and aggregate composition did not greatly vary in patient 1, we observed a reversed trend of total platelet count and PP aggregates in patient 2. In parallel with the clinical deterioration of the patient 12 days from admission, PP aggregates tremendously increased that day and were even higher on day 20. While the increased platelet consumption within the extracorporeal blood circuit presumably contributes to the observed drop of the total platelet count following initiation of ECMO therapy, the consumption of platelets on day 12 is likely caused by the formation of PP aggregates as the blood sample was collected before the start of the ECMO therapy. The decrease in platelets on day 12 is comparable to the values of day 2, where still no rapid worsening of the patient status occurred. This observation suggests that PP aggregates could be used to differentiate causes of platelet consumption.

A few limitations to this study should be mentioned. Our analysis was performed monocentric as we had only one measurement setup available, and the POC intention limited a multicentric design, resulting in a small cohort size. Although gender and age proportions were comparable for the different cohorts (Supplementary Fig. 4), a large, multicentre study with a diverse patient population is needed for reliable validation. The data were obtained over one year, from November 2020 to 2021, with patients infected with wildtype SARS-CoV-2, as well as the B.1.1.7, B.1.351, and B.1.617.2 variants. Non-hospitalized and asymptomatic SARS-CoV-2-infected individuals were not enrolled. Due to the study design, only low statistical power for LP aggregates was obtained. Therefore, LP aggregates were not further studied in this project.

In summary, we present a POC-compatible method that allows rapid, label-free phase image-based quantification of micro-aggregates. The technique was employed to characterize PP aggregates in COVID-19, and a high correlation with disease

severity was reported. While future studies are required to validate and substantiate our findings, we demonstrate that this technology yields clinically relevant findings and is not limited to COVID-19. Increased PP and LP aggregates have been observed in several medical disorders, including sepsis, cancer, or cardiovascular diseases[50–52]. This technology to study cell aggregates with single platelet resolution in low shear conditions is a promising prognostic or diagnostic tool for various systemic diseases.

## Data availability

Source Data for the main figures in the manuscript with statistical analyses are provided in Supplementary Data file 2. The medical laboratory dataset including aggregate data is not publicly available at this time due to data privacy considerations but may be available from the corresponding author on reasonable request.

## Code availability

The Mask RCNN network used in this study is available at (https://github.com/matterport/Mask_RCNN). As a backbone for the model, a ResNet50 is used (https://github.com/KaimingHe/deep-residual-networks). Morphological features were calculated using the opencv library (https://github.com/opencv/opencv). The custom code used in this work is available at https://zenodo.org/record/8427711.

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

## Acknowledgements

We are indebted to all study participants who consented to having their data published. We thank the volunteers of TranslaTUM and Klinikum München rechts der Isar for blood donations. The support from Percy Knolle, Gerhard Schneider, Peter Luppa, Ritu Mishra, and Admar Verschoor is gratefully acknowledged. This manuscript is dedicated to Professor Dr. Franz Ludwig Dickert on the occasion of his 80th birthday. S.Ro was funded by the German Federal Ministry for Education and Research (BMBF) with the funding ID ZN 01 j S17049.

## Authors contributions

S.Ra, T.L. and O.H. conceived and designed the study. S.Ra, J.E. and H.I. coordinated patient recruitment and sample collection. C.K., D.F. and D.H. performed the D.H.M. measurements and data analysis. S.Ro, M.L. and K.D. designed the image analysis. C.K., J.E., B.H., D.F. and S.Ro designed the figures and tables. S.Ra, M.S., T.L., K.D. and O.H. supervised the study. B.H. performed the statistical analysis. C.K. and J.E. drafted the initial manuscript. All authors contributed intellectual content during the drafting and revision of the work, reviewed and approved the final version.

## Funding

## Competing interests

The authors declare the following competing interests: T.L. received travel grants from Gilead, Pfizer, and M.S.D. and lecture fees from CytoSorbents. S.Ra received travel grants from Gilead and lecture fees from CytoSorbents. J.E. reports non-financial support from Gilead Sciences, ViiV Healthcare and Pfizer. All other authors declare no conflict of interest.
