## [Peer Review File · Communications Medicine]

Reviewers' comments:

Reviewer #1 (Remarks to the Author):

The authors report about a quantitative phase imaging (QPI) approach with digital holographic microscopy (DHM) for label-free assessment of platelet aggregates as biomarker for the prediction of severity in COVID-19. DHM QPI images of platelets in diluted blood of healthy donors as well as from moderate and severe COVID-19 patients are acquired utilizing a microfluidic channel with three-dimensional hydrodynamic focusing. The retrieved QPI images series are evaluated by threshold and convolutional neural network (CNN)-based segmentation to identify platelet agglomerates and their composition. The amount of platelet agglomerates as well as their composition and size distributions are correlated with the severity of COVID-19 and established biochemical markers.

In general, the manuscript is motivated, organized, and includes adequate references. The experimental investigations as well as the data evaluation appear to be accurately performed. The presented results are plausible and novel. The described label-free approach represents an advance beyond the current state-of-the-art in label-free platelet characterization and thus may be of high interest for the areas of covid-19 research, blood analysis and label-free quantitative imaging, as well as for the fields QPI and DHM. In summary, the content of the manuscript appears to be suitable for the journal Communications Medicine.

However, the authors should consider revisions:

1. Abstract: QPI with DHM/QPI-based imaging flow cytometry seems to be an essential tool for the quantitative analysis of platelet aggregations within the study. Thus, the method should be also mentioned in the abstract.

2. Introduction and discussion: In the introduction and the discussion sections, the authors should relate their approach further to the current state-of-the-art in QPI and DHM (see for example <https://doi.org/10.1021/acsnano.1c11507>, https://doi.org/10.1007/11663_2019_6, <https://doi.org/10.1016/j.optlaseng.2020.106188>, <https://doi.org/10.1021/acsnano.1c11507> and included references) and related earlier work in QPI analysis of blood cells (for instance, <https://doi.org/10.1038/s41598-017-06311-y>, <https://doi.org/10.3390/cells11040755>, <https://doi.org/10.3390/cells12050762>) and COVID-19 research <https://doi.org/10.1038/s41377-021-00620-8>.

3. Methods section:

a. It would be helpful for an interdisciplinary reader if also the other blood cell types, e.g., red blood cells, leucocytes, etc. would be indicated in figure 1b and Extended Data Fig. 7.

b. The authors should add a brief statement/explanation, why polyethylene oxide (PEO) was added to the buffer medium (this seems to be not explained within the manuscript).

4 Results section: In section "Correlation with clinical biomarker" a cross-reference to Fig. 3 seems to be missing.

5. Discussion section (major points):

a. The authors should discuss the advantages/limitations of QPI images in the study in comparison to bright field images, as shown in figure 1c, with more details.

b. The authors should discuss advantages/limitations of the approach in comparison to established flow cytometry approaches, in which, for example, forward scattering (FSC) and sideward scattering (SSC) can be applied to discriminate blood cells by size and granulation related information and, if applicable, why it is not possible to use FSC and SSC signals to detect and discriminate platelets aggregates.

Minor points:

- Figures 2-5 in the main part of the manuscript: The labelling of the plots in all figures is very tiny and should be enhanced for improved visibility.

- Extended data figure 7: a scale bar should be added to the images.

Reviewer #2 (Remarks to the Author):

The study by Klenk et al. discovered a positive relationship between the severity of COVID-19 and the presence of platelet and platelet leukocyte aggregates. Furthermore, they observed a connection between disease severity and the composition as well as the size distribution of platelet aggregates.

Major comments:

-As patients were recruited during the vaccination campaign, it should be determined whether these individuals were immunized (due to previous infections or vaccination for instance).

-It is unclear how the authors distinguish stable or moderate COVID from critical COVID?

-There is not enough information on the hospitalized cohort. Please add a table for demographic characteristics and biological data including race/ethnicity, ALT, AST, CRP, D-dimers... Also, add a second table for medication.

-Please add and discuss the following references:

34293929

37044132

35443030

35175688

Reviewer #3 (Remarks to the Author):

The manuscript by Klenk and colleagues describe a novel label free microfluidic platform as a means to detect platelet and platelet-leukocyte aggregates in whole blood. Using this, they demonstrate the correlation between platelet and platelet-leukocyte aggregates and COVID-19 disease severity.

The concept of an association between disease severity and markers of coagulation/thromboinflammation is well described. As such, the major novelty of this study lies in the technology described. On this point, although described as a potential POCT device, it is not clear how easily transferrable this technology would be to routine diagnostic laboratories? That is, the equipment

and analysis workflow?

The authors should avoid overstating their conclusions as this study has been done in a relatively small number of patients. In contrast, studies assessing CRP and COVID (although retrospective) have involved thousands of patients.

I do not quite understand the analysis of aggregate formation and COVID variants? Isn't the relevant point whether there is a correlation between clinical disease severity of COVID-19 rather than the variant. Again, the numbers for this analysis are very small.

Some discussion should be offered for why there is little to no association between aggregate no and D-dimer and CRP. These are widely available and widely studied in COVID. The lack of correlation makes one concerned that given the wide spread of data in the clinically severe cohorts, these data may not hold true in a bigger cohort.

Point-by-point response to the referees' comments

Reviewer 1:

1. Abstract: QPI with DHM/QPI-based imaging flow cytometry seems to be an essential tool for the quantitative analysis of platelet aggregations within the study. Thus, the method should be also mentioned in the abstract.

Thank you for this important comment. We agree that the type of imaging is essential and consequently have added it to the manuscript.

2. Introduction and discussion: In the introduction and the discussion sections, the authors should relate their approach further to the current state-of-the-art in QPI and DHM (see for example <https://doi.org/10.1021/acsnano.1c11507>, https://doi.org/10.1007/11663_2019_6, <https://doi.org/10.1016/j.optlaseng.2020.106188>, <https://doi.org/10.1021/acsnano.1c11507> and included references) and related earlier work in QPI analysis of blood cells (for instance, <https://doi.org/10.1038/s41598-017-06311-y>, <https://doi.org/10.3390/cells11040755>, <https://doi.org/10.3390/cells12050762>) and COVID-19 research <https://doi.org/10.1038/s41377-021-00620-8>.

Thank you for this suggestion. A consideration of QPI and QPI-related work has been added to the introduction and discussion.

3a. It would be helpful for an interdisciplinary reader if also the other blood cell types, e.g., red blood cells, leucocytes, etc. would be indicated in figure 1b and Extended Data Fig. 7.

We appreciate this comment and have added indication for platelets, erythrocytes, LP aggregates and PP aggregates in Extended Data Fig. 7b.

3b. The authors should add a brief statement/explanation, why polyethylene oxide (PEO) was added to the buffer medium (this seems to be not explained within the manuscript).

PEO is used for the viscoelastic focussing of the blood cells. This is shortly explained in the method section "Microfluidic cell alignment":

"To support focusing of platelets in the z-direction, 0.05% 2 MDa PEO is added to the solution to enable viscoelastic focusing. Preliminary tests were carried out to ensure that the addition of the polymer does not activate the platelets (Extended Data Fig. 1c and 1d)."

4. Results section: In section "Correlation with clinical biomarker" a cross-reference to Fig. 3 seems to be missing.

We apologize for being unprecise, cross-references to Fig. 3b and Fig 3c have now been added.

5a. The authors should discuss the advantages/limitations of QPI images in the study in comparison to bright field images, as shown in figure 1c, with more details.

Advantages and limitation of QPI images compared to bright filed images have been added both in the main and in the discussion section.

5b. The authors should discuss advantages/limitations of the approach in comparison to established flow cytometry approaches, in which, for example, forward scattering (FSC) and sideward scattering (SSC) can be applied to discriminate blood cells by size and granulation related information and, if applicable, why it is not possible to use FSC and SSC signals to detect and discriminate platelets aggregates.

We now elaborate on why fluorescent flow cytometry (with SSC, FSC and antibody staining) is not suitable for the application of PP aggregates in the main section. Due to the limited spatial resolution of FSC and SSC, which only gives one value per object, smaller aggregates and single cells with similar sizes are difficult to discriminate. Furthermore, POC solutions for the acute care require economic solutions and fast turn-around-time, which is difficult to achieve with a method depending on additional labeling steps.

5c. Figures 2-5 in the main part of the manuscript: The labelling of the plots in all figures is very tiny and should be enhanced for improved visibility.

We appreciate this important comment. The font size of the plots was increased and will be further adjusted for the design of the final document.

5d. Extended data figure 7: a scale bar should be added to the images.

We agree with the reviewer. The respective scale bars have been added to the images.

Reviewer 2:

1. As patients were recruited during the vaccination campaign, it should be determined whether these individuals were immunized (due to previous infections or vaccination for instance).

We thank you for this important comment. 26 of the patients had not been vaccinated at the time of the study. 8 patients reported two previous anti-SARS-CoV-2 vaccinations, one patient had been vaccinated once and in one individual the information on vaccination status was not available. The information has been added to the Extended Data Fig. 5. None of the patients reported of a previous infection, however, we cannot exclude that some patients had undergone an asymptomatic/mild infection before.

2. It is unclear how the authors distinguish stable or moderate COVID from critical COVID?

We used the WHO ordinal scale for clinical improvement for hospitalised patients with COVID-19 (doi:10.1016/s1473-3099(20)30483-7) to classify into moderate

COVID-19 (score 4 or 5) and severe COVID-19 (score 6 to 10). None of the included patients qualified for mild COVID-19 (score 0-3). The score is mentioned and referenced in the method as well as in the result section.

3. There is not enough information on the hospitalized cohort. Please add a table for demographic characteristics and biological data including race/ethnicity, ALT, AST, CRP, D-dimers... Also, add a second table for medication.

We appreciate this comment. The ethnic group, body mass index and smoking status have been added to the Extended Data Fig. 5. Additional biological data including liver enzymes, inflammatory parameters and D-dimers as well as medication are now presented in Supplementary 1.

4. Please add and discuss the following references 34293929, 37044132, 35443030, 35175688.

Thank you for bringing up these intriguing references. All of them have been included and referenced in the revised manuscript.

Reviewer 3:

1. On this point, although described as a potential POCT device, it is not clear how easily transferrable this technology would be to routine diagnostic laboratories? That is, the equipment and analysis workflow?

We believe the benefit of our technology is for bedside testing. Our laboratory prototype with a semi-automated workflow has a footprint of 30x40 cm for the imaging flow cytometer which is already comparable to the dimensions of e.g., POC blood gas analyser used 24/7 in acute care environments (not central laboratories) and matches the requirements for crowded ER, AMU, or ICU environments with limited space for routine diagnostics.

Future full workflow integration and customization will allow for even smaller footprints. From a BOM perspective, the technology is interesting for physicians as our technology is a reagent-free workflow requiring only a dilution step with a polymeric component and does not require autofocusing or other high-maintenance complexity. Repetitive testing is, therefore, feasible, which allows to study the biomarker kinetics. Our mobile systems are currently placed in clinics for further validation of cell aggregates, which again indicates the POC opportunity.

From an analysis workflow, POC applications require obviously an embedded solution with trained algorithm for close to real-time image analysis with a stand-alone PC, which was at this TRL level not implemented. This step will come when we have performed several clinical diagnostic studies for biomarker validation and improved in parallel our data workflow for a fast turn-around-time matching acute care requirements. In brief, our assessment of the technology does not indicate a major risk for POCT.

2. The authors should avoid overstating their conclusions as this study has been done in a relatively small number of patients. In contrast, studies assessing CRP and COVID (although retrospective) have involved thousands of patients.

We totally agree and for this reason we emphasize in the abstract that the technique of DHM bears the potential to establish the detection of platelet aggregates as a new biomarker. We did not evaluate in this project if this biomarker performs better or is more valid than established parameters like CRP and we do not claim that either. By nature, the first evaluation of a new biomarker will not involve thousands of patients. The sample size is limited by the fact that diagnostic devices are only available at one study site. However, we present prospective data that show promising results for platelet aggregates as a prognostic parameter. These results are the rationale for further evaluation and validation of this tool in other disease entities and in bigger patient cohorts.

3. I do not quite understand the analysis of aggregate formation and COVID variants? Isn't the relevant point whether there is a correlation between clinical disease severity of COVID-19 rather than the variant. Again, the numbers for this analysis are very small.

Indeed, our main point is the association of platelet aggregates with severity of COVID-19. But as the pathogenesis of CAC in COVID-19 is not completely understood we performed additional analyses to exclude confounding factors. The analysis of aggregate formation and SARS-CoV-2 variants supports our hypothesis of an association with severity as we observed more aggregates in the more virulent variants. We point out this aspect in our discussion.

4. Some discussion should be offered for why there is little to no association between aggregate no and D-dimer and CRP. These are widely available and widely studied in COVID. The lack of correlation makes one concerned that given the wide spread of data in the clinically severe cohorts, these data may not hold true in a bigger cohort.

Figure 3 shows a highly significant between-subject correlation of D-Dimers with platelet aggregates. This implies that high numbers of platelet aggregates correlate with high D-Dimer concentrations. However, if D-Dimers are applicable for disease monitoring is questionable. Consistently, we did not observe a significant within-subject correlation between D-Dimer and platelet aggregates. Nevertheless, a biomarker for disease monitoring would be desirable and as pointed out by our case examples PP aggregates might have this potential. However, further studies are required to investigate this aspect. With respect to the CRP, there are significantly elevated serum concentrations in patients with severe COVID-19 as compared to patients with a moderate disease (14.83 mg/dl vs 7.53 mg/dl, $p=0.05$). So, there is an association of disease severity with CRP in our cohort as well. Most likely due to a limited sample size and a wide range of absolute CRP levels in the cohort with severe COVID-19 (SD=10.1 mg/dl) the associations between platelet aggregates and CRP with COVID-19 severity did not translate into a correlation of the two biomarkers. In the revised version of our manuscript, we now present the association of CRP with severity and point out this aspect in the discussion.

REVIEWERS' COMMENTS:

Reviewer #1 (Remarks to the Author):

The authors have addressed my comments and questions.

Reviewer #2 (Remarks to the Author):

The authors have responded well to all my concerns. I think this revised version of the article deserves to be published in Communications Med.

Reviewer #3 (Remarks to the Author):

The authors have addressed my concerns.